# Antifungal Activity of Surfactin Against *Cytospora chrysosperma*

**DOI:** 10.3390/biom16010051

**Published:** 2025-12-29

**Authors:** Xinyue Wang, Liangqiang Chang, Qinggui Lian, Yejuan Du, Jiafeng Huang, Guoqiang Zhang, Zheng Liu

**Affiliations:** Xinjiang Key Laboratory of Oasis Agricultural Pest Management and Plant Protection Resources Utilization, College of Agronomy, Shihezi University, Shihezi 832003, China

**Keywords:** surfactin, pathogenic fungi, mode of action

## Abstract

*Cytospora chrysosperma* is a common opportunistically parasitic fungus that mainly infects forest trees, severely restricting the development of the fruit and forest industry. Surfactin is a secondary metabolite produced by *Bacillus* species and exhibits antifungal activity; Although the core antifungal mechanism of surfactin against plant pathogens has been extensively studied, our study found that surfactin can target the tricarboxylic acid cycle of *C. chrysosperma*. This study aimed to investigate the potential mechanism underlying the inhibitory effect of surfactin on *C. chrysosperma*. The results showed that surfactin had a significant inhibitory effect on *C. chrysosperma*, with a half-maximal effective concentration of 0.787 ± 0.045 mg/mL and a minimum inhibitory concentration of 2 mg/mL. Morphological observations revealed that surfactin significantly disrupted the morphology and ultrastructure of *C. chrysosperma* hyphae. FDA/PI staining indicated that surfactin affected the cell membrane integrity of *C. chrysosperma*, while DCFH-DA fluorescent staining and antioxidant enzyme activity assays demonstrated the accumulation of reactive oxygen species in hyphal cells following surfactin treatment. Additionally, the reduction in adenosine triphosphate content, as well as the decreased activities of ATPase and succinate dehydrogenase, suggested that energy production might be inhibited. Finally, MDC staining showed the occurrence of autophagosomes in *C. chrysosperma* hyphae after surfactin treatment, which may lead to hyphal death. Transcriptome analysis revealed that surfactin impaired the normal biosynthesis of the *C. chrysosperma* cell membrane and interfered with the tricarboxylic acid cycle by binding to citrate synthase, resulting in intracellular energy metabolism disorders. This study provides new insights into the potential mechanism by which surfactin inhibits hyphal growth of *C. chrysosperma*.

## 1. Introduction

Walnut (*Juglans regia* family Juglandaceae, genus *Juglans*) is an important economic tree species in China. Due to the numerous types of walnut diseases, their long damage cycle, and high occurrence frequency, these diseases have become major biological disasters affecting fruit yield and quality, seriously threatening the sustainable and healthy development of the fruit industry. *Cytospora chrysosperma* is a common weakly parasitic fungus that mainly affects walnut, apple, fragrant pear, and poplar [1]. This pathogen has latent infection capability, and it can cause disease throughout the year; once infections occur, it is extremely difficult to control, severely limiting forestry and fruit production [2]. Currently, disease control relies primarily on chemical fungicides, but these pose problems such as environmental pollution, safety risks, and residual toxicity [3,4]. Therefore, developing effective bio-derived pesticides as alternatives to chemical fungicides for controlling walnut canker disease has become an important strategy, offering a more sustainable approach to plant disease management. Lipopeptides, as a class of microbially derived materials with broad-spectrum antifungal activity and environmental benignity, offer a novel strategy for the green prevention and control of forest diseases—with surfactin, a typical representative of lipopeptides, being a key candidate in this context.

As secondary metabolites synthesized by microorganisms, lipopeptides possess prominent advantages including broad-spectrum antimicrobial activity, excellent stability, and low toxicity. They exhibit significant antagonistic effects against fungi, bacteria, viruses, and mycoplasmas. These unique characteristics render lipopeptides promising candidates for high-efficiency plant disease control agents, particularly demonstrating great research and application potential in combating plant diseases caused by fungal pathogens [5]. Surfactin is a representative biosurfactant in the cyclic lipopeptide family. Synthesized extracellularly under the catalysis of non-ribosomal peptide synthetases (NRPS), its chemical structure consists of a cyclic heptapeptide linked to a β-hydroxy fatty acid chain with a variable carbon chain length (containing 12–17 carbon atoms) [6]. Abdelli et al. [7] reported that surfactin exerts excellent inhibitory effects on most bacteria, including *Staphylococcus epidermidis* S61, *Staphylococcus aureus*, *Enterococcus faecium*, *Salmonella enterica*, *Escherichia coli*, and *Pseudomonas savastanoi*. In recent years, studies have demonstrated that surfactin also exhibits significant growth-inhibitory activity against various filamentous fungi, such as *Fusarium moniliforme*, *Trichoderma aggressivum* f. *europaeum*, and *Sclerotinia sclerotiorum* [8,9,10]. Chen et al. [11] found that surfactin inhibits the growth of *Fusarium graminearum* by inducing apoptotic-like cell death in its hyphae, which is mediated by the accumulation of reactive oxygen species (ROS), a decrease in mitochondrial membrane potential (MMP), activation of caspase-like activity, and chromatin condensation. Numerous studies confirm that surfactin exerts marked inhibitory effects on filamentous fungi and strong antagonism toward diverse crop phytopathogenic fungi. A key research gap, however, is that the inhibitory activity and underlying action mechanism of surfactin against *C. chrysosperma*—the causal agent of poplar valsa canker, a major forest disease—remain uncharacterized. Herein, surfactin was isolated and purified from *Bacillus velezensis* FZB42, with a structure comprising a 7-amino-acid cyclic peptide chain (L-Glu-L-Leu-D-Leu-L-Val-L-Asp-D-Leu-L-Leu) and a β-hydroxy-13-methyltetradecanoic acid chain (Figure 1). Prior research has demonstrated that this compound displays strong inhibitory effects on *C. chrysosperma*, but its antifungal mode of action is still not fully understood. To facilitate the large-scale use of surfactin in agricultural production, it is essential to further clarify the inhibitory mechanism of surfactin against plant pathogenic fungi.

This study initially investigated the antifungal activity of surfactin against *Cytospora chrysosperma*. The effect of surfactin on the hyphal morphology of *C. chrysosperma* was evaluated, and the potential antifungal mechanisms were inferred via transcriptomic analysis. Finally, the inhibitory mechanism of surfactin against *C. chrysosperma* was elucidated through the determination of a series of physiological and biochemical parameters. The results of this study provide a reference for the application of surfactin-producing microorganisms in the biological control of plant diseases.

## 2. Materials and Methods

### 2.1. Materials and Strains

Methanol was purchased from Fuyu Fine Chemical Co. (Tianjin, China). Propidium iodide (PI) solution, fluorescein diacetate (FDA), 4′,6-diamidino-2-phenylindole (DAPI), 2′,7′-dichlorofluorescin diacetate (DCFH-DA), and the cellular autophagy staining detection kit were obtained from Coolaber Technology Co. (Beijing, China). A mitochondrial membrane potential assay kit was purchased from Beijing Solarbio Science & Technology Co. (Beijing, China), and catalase (CAT) and peroxidase (POD) activity assay kits were obtained from Suzhou Grace Biotechnology Co. (Beijing, China). TransScript^®^ Uni All-in-One First-Strand cDNA Synthesis SuperMix for qPCR and PerfectStart Green qPCR SuperMix were purchased from TransGen Biotech (Beijing, China). Surfactin was extracted from the fermentation broth of *Bacillus velezensis* FZB42 by high-performance liquid chromatography (HPLC), identified by UPLC-MS, and dissolved in 0.5% DMSO for use in subsequent experiments.

*B. velezensis* FZB42 and *Cytospora chrysosperma* were preserved in the Research Team of Plant Pathogenic Microorganisms and Plant Interactions, Department of Plant Protection, Shihezi University and stored at –80 °C. For use in experiments, the strains were revived from cryopreservation and inoculated onto potato dextrose agar (PDA; 200 g potato, 20 g glucose, 15 g agar, 1 L distilled water) and lysogeny broth (LB; 10 g NaCl, 10 g tryptone, 5 g yeast extract, 1 L distilled water), respectively, with *C. chrysosperma* on PDA and FZB42 on LB-at 28 °C for 3–7 days to obtain fresh cultures.

### 2.2. Preparation of Fresh Mycelia

*Cytospora chrysosperma* was first grown on PDA until the formation of pycnidia. The pycnidia were then crushed and suspended in sterile water to prepare a 1 × 10^7^ spores/mL conidial suspension. A 100 μL aliquot of this spore suspension was inoculated into 30 mL of potato dextrose broth (PDB) and incubated with shaking at 180 r/min, 28 °C for 2 days. The resulting *C. chrysosperma* mycelia were collected, washed three times with 10 mL sterile physiological saline to obtain a fresh mycelial suspension for further experiments.

### 2.3. Antifungal Activity of Surfactin Against Cytospora chrysosperma

The antifungal activity of surfactin against *Cytospora chrysosperma* was determined using the optical density (OD) assay [12]. The spore suspension was collected following the method described in Section 2.2 and added to a 96-well plate. An equal volume of surfactin was mixed into each well to achieve final concentrations of 0, 0.03125, 0.0625, 0.125, 0.25, 0.5, 1, 2, 4, and 8 mg/mL, respectively. A 0.5% methanol solution was used as the blank control instead of surfactin, with three replicates set for each treatment group. After incubation at 28 °C for 12 h, the optical density of each well was measured at a wavelength of 600 nm using a microplate reader. The antifungal activity was characterized by the median effective concentration (EC_50_)—this value denotes the dose required to suppress mycelial growth by 50% relative to the control group, and it is calculated through concentration-response regression analysis. The minimum inhibitory concentration (MIC) of surfactin against *C. chrysosperma* was defined as the concentration at which no visible growth was observed in the wells [13].

### 2.4. Effect of Surfactin on the Morphology of C. chrysosperma

#### 2.4.1. Optical Microscope Observation

Following placement of a 5 mm mycelial block of *Cytospora chrysosperma* at the center of a PDA plate, a 5 mm circular filter paper disk was placed 2.5 cm away from the center, and 20 μL of 2 mg/mL surfactin solution was pipetted onto the disc. Finally, a sterile coverslip was inserted obliquely into the PDA medium gap between the two disks. The plate was incubated at 28 °C for 3 days, with 0.5% methanol used as a blank control instead of surfactin. The coverslip was gently removed with forceps, observed under an optical microscope, and images were captured for further evaluation.

#### 2.4.2. Scanning Electron Microscopy (SEM) Observation

Scanning electron microscopy (SEM) observation was performed according to the method previously reported by Xiang [14]. The procedure described in Section 2.4.1 was followed, except that a silicon wafer was used instead of a coverslip. Silicon wafers were gently removed with forceps to obtain hyphal samples from both the control and treatment groups. Samples were fixed in 2.5% glutaraldehyde solution for 12 h, then thoroughly rinsed with 0.1 M phosphate-buffered saline (PBS). They were subsequently treated with 1% osmium acid solution for 2 h and rinsed again with PBS. Dehydration was carried out using ethanol solutions of gradient concentrations (30%, 50%, 70%, 80%, 90%, and 100%). Samples were observed under a scanning electron microscope after critical point drying and gold sputtering.

#### 2.4.3. Transmission Electron Microscopy Observation

Transmission electron microscopy (TEM) observations were conducted following the protocol previously reported by Xu [15]. Hyphae were cultured using the solution specified in Section 2.4.2. Samples were first fixed in 25% glutaraldehyde solution, rinsed with PBS, then post-fixed in 1% osmium tetroxide solution. After another PBS rinse, samples were dehydrated via a gradient series of ethanol solutions. They were then treated with acetone before being embedded in embedding resin. Ultrathin sections were cut using an ultramicrotome, stained sequentially with uranyl acetate and lead citrate solutions, and finally observed under a transmission electron microscope.

### 2.5. Effect of Surfactin on Cell Membrane Integrity of C. chrysosperma

A dual fluorescence staining with fluorescein diacetate (FDA) and propidium iodide (PI) was used to detect the cell viability of *Cytospora chrysosperma* [16]. *C. chrysosperma* hyphae were collected as described in Section 2.2. Then 20 μL of FDA solution (5 mg/mL) and 1 μL of PI solution (1 mg/mL) were added to the hyphae. The mixture was incubated at room temperature in the dark for 8 min. After staining, the hyphae were washed three times with sterile physiological saline, resuspended, and prepared into slides. Cell viability was observed under a laser confocal scanning microscope. The optimal excitation and emission wavelengths of FDA were 488 nm and 530 nm, respectively. For PI, the optimal excitation and emission wavelengths were 535 nm and 615 nm, respectively.

### 2.6. Effect of Surfactin on Oxidative Stress in C. chrysosperma

The intracellular reactive oxygen species (ROS) production was detected using the DCFH-DA fluorescence assay according to the method described by Duan et al. [17]. *C. chrysosperma* hyphae were collected as detailed in Section 2.5. A 1 mL aliquot of 10 μmol/mL DCFH-DA solution was added to the hyphae, followed by incubation in the dark for 20 min. After staining, the hyphae were washed three times with sterile physiological saline, resuspended, and prepared into slides. Observations and photographs were taken under a fluorescence microscope at an excitation wavelength of 488 nm. The subsequent determination of SOD, CAT and POD contents were carried out in strict accordance with the instructions of the corresponding detection kits.

### 2.7. Mitochondrial Membrane of C. chrysosperma

JC-1, an anthocyanidin dye, exhibits green fluorescence in its monomeric form under normal conditions. However, when mitochondrial membrane potential is high, this dye accumulates in metabolically active mitochondria and assembles into aggregates that emit red fluorescence [18]. *C. chrysosperma* hyphae were collected using the protocol described in Section 2.5. A 0.5 mL volume of JC-1 staining working solution was added to the hyphae, and the mixture was gently inverted multiple times to ensure uniform mixing, then incubated in the dark for 20 min. Post-staining, the hyphae were washed three times with sterile physiological saline, resuspended, and prepared into slides. Observations and imaging were performed using a laser confocal scanning microscope at excitation wavelengths of 490 nm and 525 nm, with simultaneous detection of fluorescence intensity.

### 2.8. Effect of Surfactin on Energy Metabolism of C. chrysosperma

Mycelial culture was performed as described in Section 2.2. Under sterile conditions, 2 mg/mL surfactin solution was added to the mycelia; meanwhile, a blank control group treated with 0.5% dimethyl sulfoxide (DMSO) was set up. After 4 h of treatment, the mycelia were harvested, rinsed with phosphate-buffered saline (PBS), and subjected to vacuum filtration. Subsequent determinations of ATP content, ATPase activity, and succinate dehydrogenase activity were strictly conducted following the instructions of the respective detection kits.

### 2.9. Observation of Autophagy in C. chrysosperma Treated with Surfactin

Autophagy induction in hyphae was detected using monodansylcadaverine (MDC) fluorescent staining. Hyphae were cultured following the method described in Section 2.5. A 200 μL aliquot of MDC working solution was added, and the mixture was stained at room temperature in the dark for 30 min. After staining, the supernatant was discarded, and the hyphae were washed three times with 1× wash buffer. Observations and imaging were then performed under a laser confocal scanning microscope.

### 2.10. Effect of Surfactin on Transcriptome of C. chrysosperma

To investigate the transcriptomic response of *C. chrysosperma* following surfactin treatment, the hyphae for analysis were harvested as detailed in Section 2.2. A 2 mg/mL surfactin solution was added to the hyphae, while 0.5% methanol was used as the blank control instead of surfactin, with each treatment performed in triplicate. Following 4 h of treatment, the hyphae were washed with PBS buffer and quickly frozen in liquid nitrogen for preservation. Procedures including RNA extraction, purification, reverse transcription, library construction, and sequencing were entrusted to Nanjing Personal Gene Technology Co., Ltd. (Nanjing, China). Kyoto Encyclopedia of Genes and Genomes (KEGG) was employed for significant enrichment analysis to identify the metabolic pathways involving the differentially expressed genes in *C. chrysosperma* under surfactin treatment.

### 2.11. RT-qPCR Analysis

To confirm the reliability of the RNA-seq dataset, five genes were chosen for verification via RT-qPCR analysis. Hyphae were pretreated as described in Section 2.10. Reverse transcription was performed using TransScript^®^ Uni All-in-One First-Strand cDNA Synthesis SuperMix for qPCR. RT-qPCR was conducted with PerfectStart Green qPCR SuperMix. The primers utilized in this work are detailed in Table 1. Actin served as the internal reference gene for normalizing cDNA concentrations, and the relative expression of target genes was determined via the 2^−ΔΔCt^ method [16].

### 2.12. Molecular Docking

The amino acid sequence of citrate synthase from *C. chrysosperma* was derived from the nucleotide sequence of its core gene g632, and this sequence was uploaded to the SWISS-MODEL platform (https://swissmodel.expasy.org) to perform homology modeling. For this process, the protein sequence of a characterized crystal structure was used as the modeling template; since it exhibited high crystal structure homology with the *C. chrysosperma* citrate synthase, it was chosen as the template for homology modeling. Subsequently, molecular docking of the *C. chrysosperma* citrate synthase protein model with two small molecules (surfactin and oxaloacetate) was carried out via the CB-DOCK2 platform (http://cadd.labshare.cn/cb-dock2/, accessed on 25 November 2025). Furthermore, Discovery Studio (released in 2021) was employed to characterize the binding interactions between these small molecules and the target protein.

### 2.13. Statistical Analysis

To evaluate the significance of differences between mean values (at the threshold of *p* < 0.05), one-way analysis of variance (ANOVA), Duncan’s multiple range test, and Student’s t-test were employed, with at least three replicates set for each condition (*n* = 3). All statistical calculations were conducted via SPSS 26.0 software, while experimental graphs were plotted using Origin 2024 software.

## 3. Results

### 3.1. Antifungal Activity of Surfactin Against C. chrysosperma

This study evaluated the antifungal activity of surfactin against *C. chrysosperma*. The results showed that surfactin significantly inhibited the growth of *C. chrysosperma*, with a median effective concentration (EC_50_) of 0.787 ± 0.045 mg/mL. Meanwhile, the minimum inhibitory concentration (MIC) was determined to be 2 mg/mL. In conclusion, the above results indicate that surfactin exhibits good inhibitory activity against *C. chrysosperma*. To better observe the effect of surfactin on *C. chrysosperma*, the MIC value will be adopted in subsequent experiments (Figure 2).

### 3.2. Effect of Surfactin on the Morphology of C. chrysosperma

We assessed the impacts of surfactin on the morphological structure of *C. chrysosperma* mycelia via optical microscopy, scanning electron microscopy (SEM), and transmission electron microscopy (TEM). Under the optical microscope, mycelia in the control group presented a smooth surface, consistent thickness, and clear branching patterns (Figure 3A). In contrast, mycelia in the surfactin-treated group displayed notable morphological alterations: their surfaces turned rough, mycelia aggregated, and branching growth was suppressed (Figure 3B,C). Further SEM observations revealed that control-group mycelia had uniform thickness and a smooth surface (Figure 3D), while treated-group mycelia showed abnormal morphological traits-including surface wrinkles, depressions, and distortions (Figure 3E,F). Observation of the mycelial ultrastructure under the TEM showed that the cell walls and cell membranes of the control group mycelia were uniformly thick, with no extracellular exudates, and organelles were evenly distributed throughout the cytoplasm (Figure 3G). However, the cell walls and cell membranes of *C. chrysosperma* mycelia treated with surfactin were irregular and wavy, with unevenly distributed organelles (Figure 3H,I). Therefore, surfactin treatment can damage the cell walls and cell membranes of *C. chrysosperma* mycelia, thereby altering the mycelial morphology and ultrastructure, leading to mycelial malformation and cell necrosis.

### 3.3. Effects of Surfactin on Cell Membrane Integrity and Viability

To evaluate whether surfactin affects the cell membrane integrity of *C. chrysosperma*, we performed FDA/PI double staining of the hyphae after surfactin treatment. FDA is a non-fluorescent substrate that can diffuse into live cells and is hydrolyzed by intracellular esterases to produce fluorescent products, indicating viable cells, whereas PI is a red-fluorescent nucleic acid stain that can enter cells only when the cell membrane is severely damaged (it labels non-viable, membrane-compromised cells). As shown in the fluorescence images, surfactin-treated *C. chrysosperma* hyphae displayed a large amount of red fluorescence compared to the control (which showed predominantly green fluorescence from FDA) (Figure 4). The abundance of red-stained (PI-positive) hyphae in the treated group indicates that surfactin treatment led to extensive loss of membrane integrity and cell death in the fungal mycelium. This result demonstrates that surfactin significantly compromises the viability of *C. chrysosperma* cells.

### 3.4. Effects of Surfactin on Oxidative Stress in C. chrysosperma

ROS, as key oxidative damage factors, can cause cellular dysfunction and even cell death [19]. The DCFH-DA staining method was used to detect the oxidative stress level in *C. chrysosperma* hyphae after surfactin treatment. This probe is cell-permeable and non-fluorescent in its native state; it is enzymatically hydrolyzed to DCFH inside cells, and DCFH is easily oxidized to DCF (a green fluorescent compound) in the presence of ROS. As shown in the figure, compared with the blank control, the fluorescence intensity of the treated hyphae was significantly enhanced, indicating that surfactin can induce oxidative damage in *C. chrysosperma*, thereby leading to the accumulation of ROS in hyphae (Figure 5A). SOD, CAT, and POD are key antioxidant enzymes in fungi that work synergistically to maintain fungal redox homeostasis and enhance stress resistance [20]. This study determined the effect of surfactin treatment on the antioxidant enzymes of *C. chrysosperma*. After 4 h of surfactin treatment, compared with the SOD, CAT, and POD activities of the control group, the enzyme activities of the ½MIC and MIC treatment groups were significantly reduced (Figure 5B–D). The decrease in the activities of these three antioxidant enzymes indicates that surfactin may inhibit or damage the fungal antioxidant enzyme system, thereby exacerbating oxidative stress. Taken together, these results demonstrate that surfactin can induce oxidative stress in *C. chrysosperma* (evidenced by increased ROS accumulation) while disrupting its antioxidant defense capacity. The accumulation of ROS and the resulting oxidative damage may be one of the important mechanisms by which surfactin inhibits the growth of this fungus.

### 3.5. Effect of Surfactin on Mitochondrial Membrane Potential

Mitochondrial membrane potential (MMP) is an important indicator of cellular health and functional status [21]. Mitochondria in the control group exhibited a polarized state with elevated membrane potential (Figure 6). By comparison, hyphae treated with surfactin displayed predominantly green fluorescence (corresponding to JC-1 monomers), alongside faint red fluorescence signals-a sign that mitochondrial membrane potential (MMP) had collapsed (Figure 6). Put simply, surfactin led to a marked decrease in the MMP of *C. chrysosperma* hyphae, which serves as a key indicator of mitochondrial dysfunction or depolarization. Diminished MMP undermines ATP synthesis and is frequently linked to the activation of programmed cell death pathways. The surfactin-mediated MMP depolarization identified in this study implies that surfactin impairs mitochondrial function in *C. chrysosperma* by inducing oxidative damage to mitochondria (via ROS) and directly disrupting the mitochondrial membrane structure.

### 3.6. Effects of Surfactin on Energy Metabolism

Mitochondria are the central hubs of cellular metabolism, integrating numerous anabolic and catabolic processes, and energy metabolism is critical for cell viability. In fungi, inhibition of energy production can impede growth, cause abnormal morphogenesis and physiology, and ultimately lead to cell death [22]. Intracellular ATP content and ATPase activity serve as indicators of the cellular energy state [23]. We therefore measured the ATP content and ATPase activity in *C. chrysosperma* hyphae after surfactin treatment. After 4 h of exposure to surfactin, the ATP levels in the hyphae dropped markedly. The control hyphae had an ATP content of 0.4058 μmol/g, whereas the 1 mg/mL (½MIC) and 2 mg/mL (MIC) surfactin-treated groups had ATP contents of only 0.1436 μmol/g and 0.0847 μmol/g, respectively (Figure 7A). Similarly, the ATPase activity in treated hyphae was significantly reduced: the control showed ATPase activity of 10.47 U/g, which decreased to 4.06 U/g at 1 mg/mL and 3.48 U/g at 2 mg/mL surfactin (Figure 7B). We also measured the activity of succinate dehydrogenase (SDH), a key enzyme in the tricarboxylic acid cycle and electron transport chain. SDH activity in control hyphae was 44.46 U/g, and this dropped to 17.90 U/g and 15.30 U/g in the ½MIC and MIC treatment groups, respectively, after 4 h (Figure 7C). These data demonstrate that surfactin disrupts the energy metabolism of *C. chrysosperma*: the fungus’s ability to produce ATP is hampered, likely due to both impaired mitochondrial function (as evidenced by loss of MMP) and direct enzyme inhibition, leading to a collapse in energy generation.

### 3.7. Surfactin Induces Autophagy in C. chrysosperma

Autophagy is a unique cellular process in eukaryotic cells. Its core function is to degrade and recycle the components in the cytoplasm, which is usually initiated when cells encounter adverse conditions such as stress or damage. [24]. We investigated whether surfactin triggers autophagy in *C. chrysosperma* using MDC staining, which labels autophagic vacuoles with punctate fluorescenc. In control hyphae, MDC staining showed a relatively uniform, diffuse fluorescence, indicating basal or no significant autophagy. In contrast, surfactin-treated hyphae exhibited clear signs of autophagy: we observed chromatin condensation and nuclear fragmentation, along with numerous bright green fluorescent puncta distributed throughout the cells (Figure 8). The overall fluorescence intensity was also enhanced in treated hyphae. The punctate MDC staining pattern in surfactin-treated hyphae corresponds to the accumulation of autophagic vesicles. These results demonstrate that surfactin induces autophagy in *C. chrysosperma*. The onset of autophagy in the treated fungal cells likely reflects a cellular response to the severe internal damage inflicted by surfactin, as the cell attempts to recycle damaged components and maintain homeostasis.

### 3.8. Transcriptomic Response of C. chrysosperma to Surfactin

To explore the molecular mechanism underlying surfactin’s activity against *C. chrysosperma*, RNA-Seq analysis was conducted. The PCA scatter plot revealed a clear separation between the 2 μg/mL surfactin-treated group and the control group (Figure 9A), confirming that surfactin exerts a distinct impact on *C. chrysosperma*. A total of 4332 differentially expressed genes (DEGs) were identified in *C. chrysosperma* mycelia (Figure 9B), including 1522 up-regulated genes and 2810 down-regulated genes. KEGG enrichment analysis showed that the significantly enriched pathways included: the ubiquitin pathway (linked to protein degradation and tumorigenesis), the steroid biosynthesis pathway (critical for cell membrane structure), the glycolysis-gluconeogenesis pathway (core of glucose metabolism and energy production), the glutathione metabolism pathway (key for cellular redox homeostasis and detoxification), and the amino sugar and nucleotide sugar metabolism pathway (required for cell wall and membrane precursor synthesis) (Figure 9C). Notably, multiple genes in the steroid biosynthesis and fatty acid biosynthesis pathways were markedly down-regulated (Figure 9E), such as sterol dehydrogenase (EC 1.3.1.72), C45 sterol isomerase (ERG2), sterol C-24 reductase (DWF1), and cytochrome P450 monooxygenase (EC 1.14.18.10). This down-regulation of core genes related to membrane biosynthesis was highly consistent with the TEM phenotypic observations—“blurred” cell wall and membrane boundaries in surfactin-treated *C. chrysosperma* mycelia-implying that surfactin may impair the integrity of *C. chrysosperma* cell walls and membranes by suppressing the expression of structure-related genes.

In addition, among the 11 genes enriched in the TCA cycle, 10 were significantly up-regulated, while only the citrate synthase-encoding gene (g632) was significantly down-regulated (Figure 9F). This expression pattern may interfere with subsequent energy metabolism processes in *C. chrysosperma*. Consistent with physiological index assays, the expression changes in genes related to energy metabolism pathways in the transcriptome were highly aligned with the significant reductions in ATP content, ATPase activity, and succinate dehydrogenase activity in treated mycelia. This suggests that surfactin treatment may impede energy production in mycelia by suppressing the expression of key genes in these metabolic pathways. In the mitophagy-yeast pathway (ko04139), 15 genes were significantly up-regulated; additionally, 34 genes were significantly up-regulated in the autophagy-yeast pathway (ko04138). Consistent with MDC staining results, the up-regulation of autophagy-related genes in the transcriptome coincided with the occurrence of autophagosomes in treated mycelia, reflecting severe disruption of intracellular homeostasis, which necessitates the activation of autophagy to clear damaged components. Of the 5 genes analyzed by qPCR, 1 was detected to be up-regulated and 1 was down-regulated (Figure 9D). This is consistent with the findings from RNA-Seq, thus confirming the accuracy of the RNA-Seq data.

These results indicate that surfactin has the potential to affect cell wall and membrane integrity, energy metabolism, mitochondrial function, and cellular homeostasis in *C. chrysosperma*.

### 3.9. Molecular Docking

To clarify the potential binding ability of surfactin to citrate synthase (CS), molecular docking technology was employed to verify this hypothesis. We performed molecular simulation docking of surfactin and oxaloacetate with CS from *C. chrysosperma* using the CB-DOCK2 platform, aiming to explore the key interacting residues in the binding pocket and differences in binding affinity between the ligands and receptor. The results showed that surfactin and oxaloacetate bound to similar pockets of the *C. chrysosperma* CS protein model (Figure 10A). Surfactin formed hydrogen bond interactions with CS residues ALA336, ARG61, ILE63, and LEU66, which differed from the binding site of oxaloacetate with CS (Figure 10B). The Vina score of surfactin-CS binding was −7.0, indicating a stronger binding affinity than that of oxaloacetate-CS (−4.4). It is inferred that the large molecular structure of surfactin occupies the native binding pocket of oxaloacetate on CS after binding, thereby preventing CS from catalyzing the condensation of oxaloacetate and acetyl-CoA to form citrate, ultimately disrupting the TCA cycle of the pathogen.

## 4. Discussion

Surfactin is an amphiphilic cyclic lipopeptide molecule [25] that acts as an anti-adhesive agent to inhibit biofilm formation, thereby influencing various host-microbe interactions [26]. Its antimicrobial efficacy depends on both the composition of the microbial cell membrane and the ratio of surfactin to membrane components [27]. Chen et al. [11] found that surfactin exerts distinct effects on *F. graminearum* hyphae depending on its concentration: at a high dose (1000 μg/mL), it damages the hyphal membrane and triggers necrosis; in contrast, at a low concentration (100 μg/mL), it fails to disrupt membrane integrity but induces apoptosis. Notably, filamentous fungi—characterized by ergosterol-rich membranes and extracellular matrices—are less sensitive to surfactin compared to bacteria. In recent years, growing research has confirmed surfactin’s critical role in controlling plant fungal diseases. For example, Chowdhury et al. observed that when *Rhizoctonia solani* infects cruciferous crops, *Bacillus amyloliquefaciens* FZB42 secretes large amounts of surfactin in lettuce rhizosphere. A comparison showed that, unlike the wild-type strain FZB42, the surfactin-deficient mutant of *B. amyloliquefaciens* FZB42 could not reduce lettuce disease incidence, proving surfactin’s key role in disease suppression [28]. Additionally, Krishnan et al. discovered that surfactin prevents *F. moniliforme* contamination in maize grains: it not only ensures normal maize seed germination but also alleviates mycotoxicosis induced by this fungus in animal models. In this study, the MIC of surfactin against *C. chrysosperma* was measured as 2 μg/mL, suggesting its potential antifungal activity against this pathogen.

We also hypothesize that surfactin may have additional antifungal mechanisms (beyond those discussed in existing studies). For instance, Krishnan et al. found that surfactin inhibits *F. moniliforme* growth and damages its hyphae in vitro by interfering with the fungus’s DNA and proteins, while reducing its glutathione content [29]. This study confirmed that surfactin can induce apoptosis in *C. chrysosperma* cells. To verify whether surfactin can induce apoptosis in the fungal hyphae, morphological observations via microscopy and physiological and biochemical index detections were performed. The results showed that surfactin treatment leads to increased cell membrane permeability, ROS accumulation, MMP reduction, and autophagosome formation in *C. chrysosperma* hyphae. These findings clearly indicate that surfactin can induce the death of *C. chrysosperma* hyphae.

Existing studies have shown that ROS can induce various biological processes, including apoptosis. Zhu et al. found that harmane, an extract from *Peganum harmala* seeds, increases cell mortality by inducing intracellular ROS accumulation in *Fusarium oxysporum* hyphae [30]. Ito et al. reported that α-tomatine exerts fungicidal effects by inducing rapid ROS-mediated programmed cell death [31]. In most eukaryotic cells, ROS are mainly produced in mitochondria, and ROS accumulation may be triggered by changes in MMP [32]. A normal MMP plays an important role in ATP production and is also crucial for maintaining mitochondrial function [33]. Chiloscyphenol A, a natural small molecule isolated from Chinese liverworts, exerts a bactericidal effect against *Candida albicans* via a dual mechanism that simultaneously triggers mitochondrial dysfunction and disrupts cell membrane integrity [34]. In this study, it was similarly found that surfactin reduces MMP and affects the normal energy metabolism of *C. chrysosperma* hyphae. Citrate synthase (CS) has been identified as a critical target for inhibiting organismal growth in previous studies. As a key mitochondrial enzyme, CS catalyzes the condensation of acetyl-CoA and oxaloacetate into citrate within the mitochondrial matrix; this reaction participates in energy production via the TCA cycle and links to the electron transport chain. Huang et al. reported that 4-(Diethylamino)salicylaldehyde could competitively inhibit CS, disrupting cellular energy generation and ultimately inducing apoptosis in *R. solani* cells [35]. Ren et al. demonstrated that hypersuccinylation of CS impairs its enzymatic function, which further suppresses the proliferation and migration of colon cancer cells [36]. Hu et al. found that *Bombyx mori* nucleopolyhedrovirus (BmNPV) infection upregulates CS acetylation in *B. mori*, causing intracellular energy metabolism dysfunction and consequently inhibiting viral self-proliferation [37]. Consistently, transcriptomic analysis in this study revealed that the gene (g632) encoding CS in *C. chrysosperma* (treated with surfactin) was significantly downregulated in the TCA cycle. Additionally, molecular docking simulations showed that surfactin exhibits stronger binding affinity to CS than oxaloacetate. These results suggest that surfactin may interfere with energy metabolism and survival of *C. chrysosperma* by binding to CS. However, only one surfactin isomer was investigated in this study. Further research is required to determine whether other surfactin isomers target CS through the same mechanism.

Autophagy is a cellular process that maintains intracellular homeostasis by degrading and recycling proteins and organelles. Under normal conditions, autophagy exerts a protective effect on cells; however, disruption of autophagic machinery or excessive activation of autophagic flux often leads to cell death. Kulkarni et al. proposed that an attractive therapeutic strategy is to regulate the intrinsic cell death pathways of fungi through drugs to induce fungal death. Targeting the inherent cell death programs of fungi may help develop antifungal drugs with rapid onset and pathogen specificity [38]. Wang et al. found that *Streptomyces* secretes rapamycin, which alters the acetylome of fungi and thereby significantly induces autophagy in *F. graminearum* [39]. Struyfs reported that a moderate dose of HsAFP1 can induce autophagy in *Saccharomyces cerevisiae* [40]. In our study, autophagy was also observed in hyphae treated with surfactin, and transcriptomic data analysis showed that genes related to the autophagy pathway were significantly up-regulated. This may be due to surfactin accelerating the self-degradation of cellular structures, leading to hyphal death.

As one of the most effective biosurfactants known to date, surfactin not only exhibits broad-spectrum antibacterial, antiviral, and antifungal activities but also serves as a high-efficiency stabilizer, emulsifier, and surface modifier, thereby expanding its applications in the food industry. From the perspective of core performance, most microbially derived surfactants hold significant advantages over synthetic counterparts, characterized by lower surface tension, interfacial tension, and critical micelle concentration (CMC), as well as superior application efficiency. Typical products can reduce the surface tension of water from 72 mN/m to 30–35 mN/m and the water-oil interfacial tension from 40 mN/m to 1 mN/m [41]. Due to their low CMC values, biosurfactants spontaneously aggregate into nanosized micelles when their concentration exceeds the threshold. The unique structure of these micelles—with a hydrophobic core and a hydrophilic surface—enables them to effectively encapsulate water-insoluble substances (e.g., chemicals, functional molecules), significantly enhancing the transport and retention of these substances in the aqueous phase and thereby improving the bioavailability of target compounds [42,43]. In terms of environmental safety, biosurfactants are derived from microbial fermentation, exhibiting lower toxicity to aquatic flora and fauna and greater susceptibility to degradation by microorganisms in soil and aquatic environments [44]. They not only improve soil quality and promote plant growth but also enhance the biodegradation of pollutants and chemical pesticides in agricultural soils [42,45,46,47]. Notably, with its excellent antifungal activity, surfactin demonstrates promising inhibitory potential against the forest pathogenic fungus *C. chrysosperma*, making it a potential alternative to traditional chemical agents as a core formulation for the green prevention and control of forest diseases. In contrast, chemical control remains the primary method for forest disease management; however, chemical agents generally pose unresolved issues such as environmental pollution, risks to human and animal safety, and pesticide residues. Although microbially derived surfactants meet the requirements of green prevention and control, they face two core constraints: first, their production cost is higher than that of synthetic surfactants; second, numerous factors during fermentation interfere with microbial growth and metabolism, hindering the high-yield production of biosurfactants. Based on this, screening low-cost substrates for microbial surfactin synthesis and optimizing its production process are of great practical significance for unlocking the application potential of surfactin in inhibiting *C. chrysosperma* and addressing the predicament of chemical control in forest disease management.

In summary, based on the above findings, a preliminary model is proposed to elucidate the inhibitory mechanism of surfactin against *C. chrysosperma* mycelia: Under surfactin treatment, the mycelial cell membrane of *C. chrysosperma* is damaged, but this damage is insufficient to induce necrosis. Accumulation of a large amount of ROS triggers a series of cellular events, including mitochondrial membrane damage. Mitochondria impaired by ROS tend to produce more ROS, leading to exacerbated oxidative stress, which in turn induces mitochondrial dysfunction characterized by decreased mitochondrial membrane potential. Additionally, surfactin binds to citrate synthase and interferes with the TCA cycle, resulting in intracellular energy metabolism disorders. These findings deepen the understanding of the antifungal mechanism of surfactin, identify and predict CS as the target of surfactin in antagonizing *C. chrysosperma*, and highlight the application potential of surfactin as an antifungal agent. Furthermore, this study lays a foundation for the development of surfactin-based antifungal agents against *C. chrysosperma*.

## 5. Conclusions

In conclusion, surfactin exhibits significant antifungal activity against *C. chrysosperma*, with a EC_50_ of 0.787 ± 0.045 mg/mL and a MIC of 2 mg/mL. It can severely disrupt the ultrastructure of *C. chrysosperma* and inhibit mycelial growth. Comprehensive analyses, including enzyme activity assays and transcriptome sequencing, reveal that the antifungal mechanisms of surfactin mainly involve three aspects: (1) causing cell membrane damage to trigger ROS accumulation; (2) impairing mitochondrial membrane potential, leading to exacerbated oxidative stress; and (3) binding to citrate synthase to interfere with the TCA cycle, resulting in intracellular energy metabolism disorders. All three mechanisms are capable of inhibiting mycelial growth or inducing cell death. These findings collectively highlight the antagonistic activity of surfactin against *C. chrysosperma*, and CS can be further explored as a target for the development of surfactin-based antifungal agents against this pathogen.

## Figures and Tables

**Figure 1 biomolecules-16-00051-f001:**
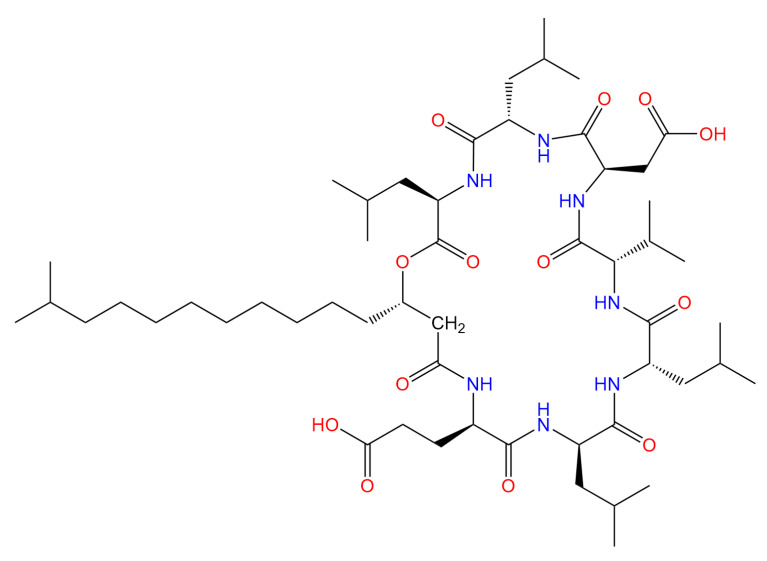
Chemical structure of surfactin.

**Figure 2 biomolecules-16-00051-f002:**
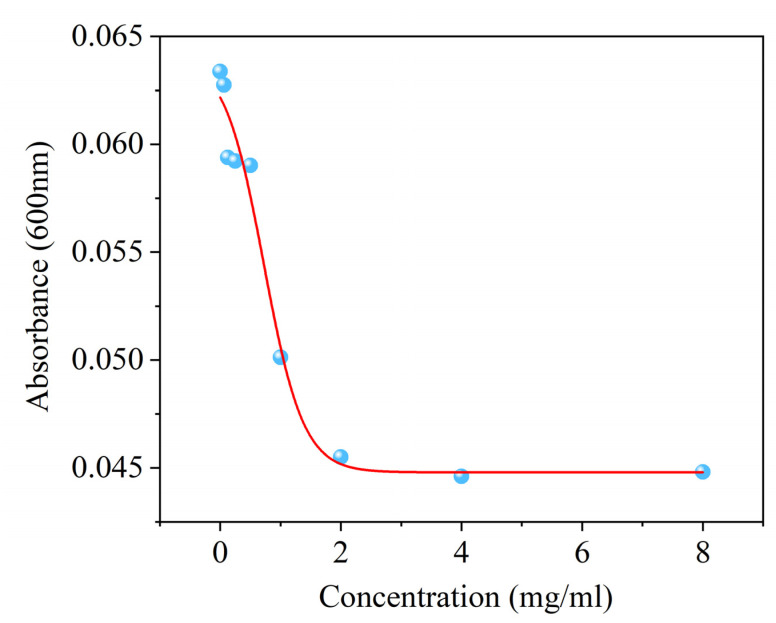
Effect of surfactin on the mycelial growth of *C. chrysosperma*.

**Figure 3 biomolecules-16-00051-f003:**
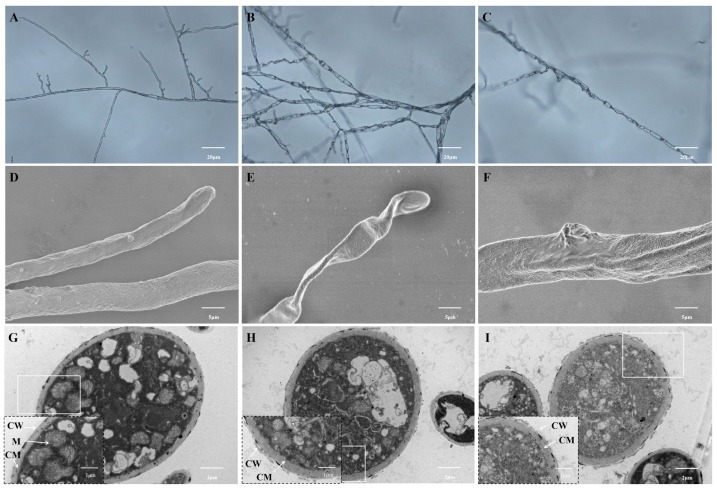
Microscopic observation of hyphal morphology of *C. chrysosperma*. Hyphae treated with 0 mg/mL surfactin (control) (**A**,**D**,**G**) and 2 mg/mL surfactin (**B**,**C**,**E**,**F**,**H**,**I**) were observed. (**A**–**C**) Optical microscopy images of hyphae (with coverslip culture method). (**D**–**F**) Scanning electron micrographs of hyphae. (**G**–**I**) Transmission electron micrographs of hyphal ultrathin sections. CW: cell wall; CM: cell membrane; M: mitochondria.

**Figure 4 biomolecules-16-00051-f004:**
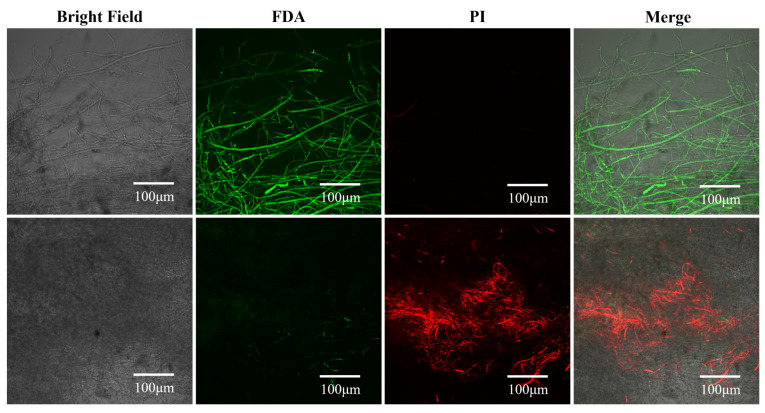
Effects of surfactin on the cell membrane integrity of *C. chrysosperma* (FDA/PI double staining). Surfactin-treated hyphae show extensive red fluorescence (PI) compared to control hyphae, indicating loss of membrane integrity and cell viability. In contrast, the control hyphae exhibit obvious green fluorescence (FDA), which reflects intact cell membranes and high cell viability.

**Figure 5 biomolecules-16-00051-f005:**
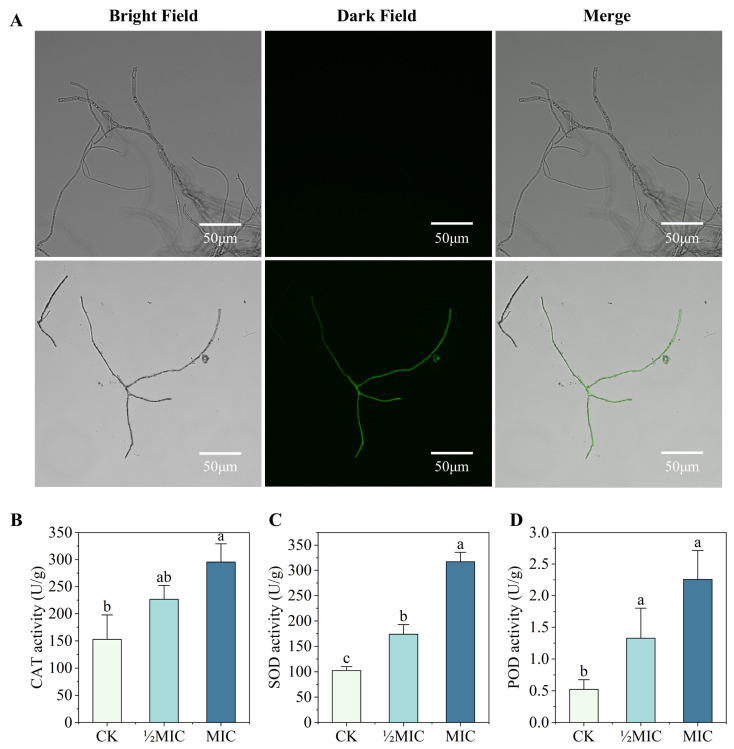
Effects of surfactin treatment on oxidative stress in *C. chrysosperma*. (**A**) DCFH-DA staining of hyphae showing ROS levels (green fluorescence); the surfactin-treated hyphae exhibit higher fluorescence intensity than the control, indicating ROS accumulation. (**B**) CAT activity in control vs. surfactin-treated hyphae. (**C**) SOD activity. (**D**) POD activity. Enzyme activities were significantly lower in surfactin-treated samples (1 mg/mL and 2 mg/mL) than in the control. Different lowercase letters indicated that there were significant differences between different treatments (*p* < 0.05).

**Figure 6 biomolecules-16-00051-f006:**
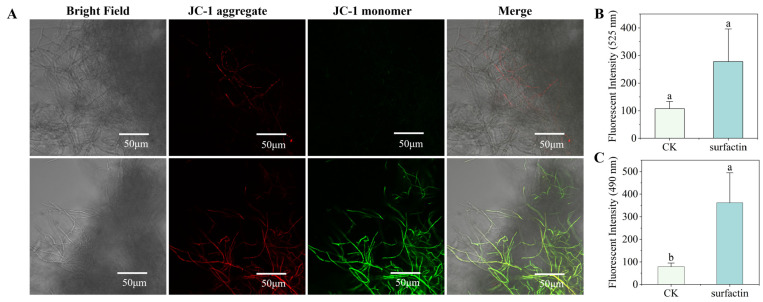
The effect of surfactin on mitochondrial membrane potential of *C. chrysosperma*. (**A**) JC-1 staining (red fluorescence represents JC-1 aggregates with normal mitochondrial membrane potential; green fluorescence represents JC-1 monomers with decreased mitochondrial membrane potential). (**B**) JC-1 aggregate fluorescence intensity statistics. (**C**) JC-1 monomer fluorescence intensity statistics. Different lowercase letters indicated that there were significant differences between different treatments (*p* < 0.05).

**Figure 7 biomolecules-16-00051-f007:**
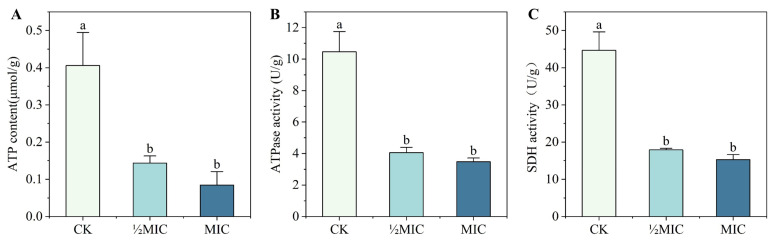
Effects of surfactin on energy metabolism in *C. chrysosperma*. (**A**) Intracellular ATP content in control and surfactin-treated hyphae. (**B**) ATPase activity. (**C**) SDH activity. All measured parameters (ATP levels and enzyme activities) were substantially lower in hyphae treated with 1 mg/mL or 2 mg/mL surfactin compared to the control. Different lowercase letters indicated that there were significant differences between different treatments (*p* < 0.05).

**Figure 8 biomolecules-16-00051-f008:**
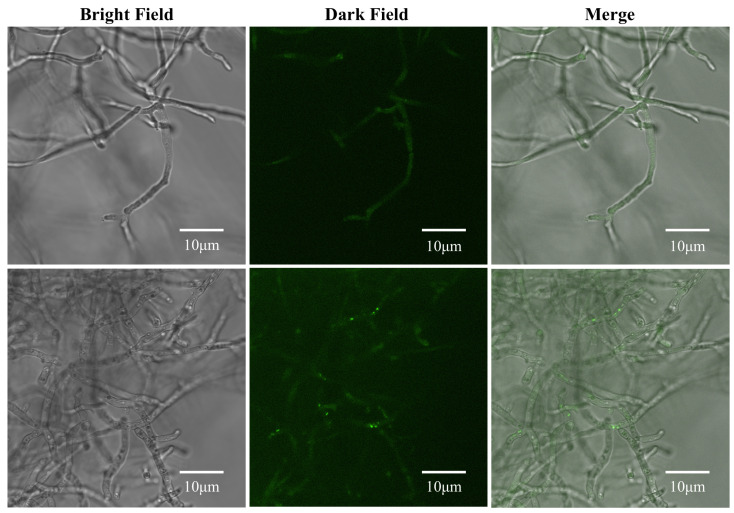
Effect of surfactin on autophagy in *C. chrysosperma*. MDC staining of hyphae shows diffuse green fluorescence in control cells, whereas surfactin-treated hyphae display intense punctate green fluorescent spots and fragmented nuclei, indicating the formation of autophagic vacuoles.

**Figure 9 biomolecules-16-00051-f009:**
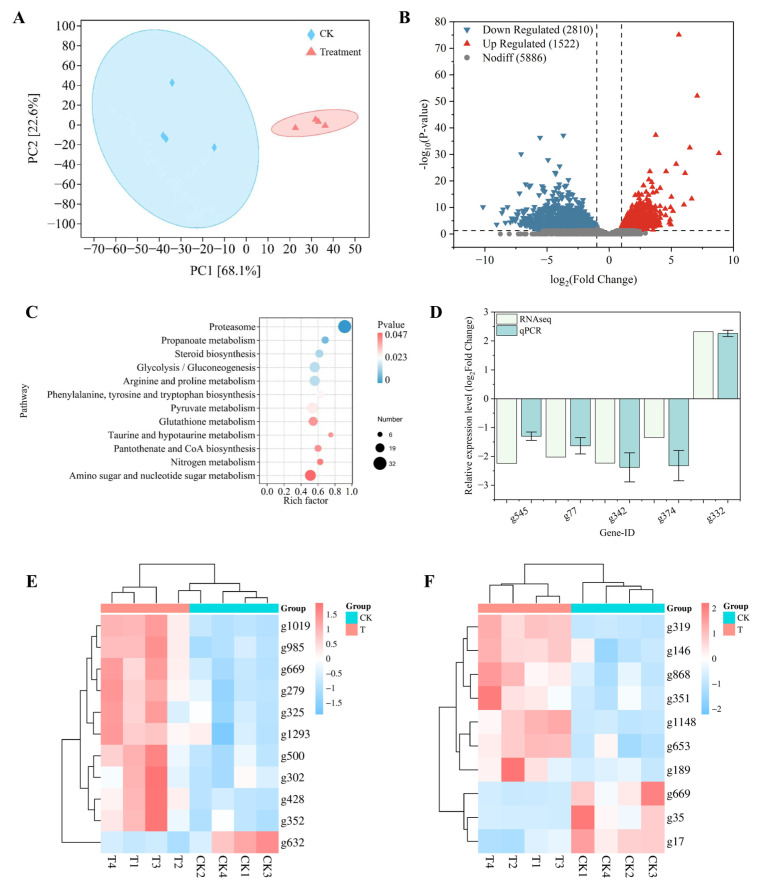
Transcriptomic and RT-qPCR analyses of *C. chrysosperma* under surfactin treatment. (**A**) Principal component analysis of gene expression profiles, showing distinct clustering of surfactin-treated vs. control samples. (**B**) Volcano plot of DEGs induced by surfactin (red dots: up-regulated genes; blue dots: down-regulated genes). (**C**) KEGG pathway enrichment analysis of major DEGs. (**D**) Verification of the RNA-seq accuracy by RT-qPCR. (**E**) Gene Expression Heatmap of Fatty Acid Biosynthesis and Steroid Biosynthesis Pathways. (**F**) TCA Cycle Pathway Gene Expression Heatmap.

**Figure 10 biomolecules-16-00051-f010:**
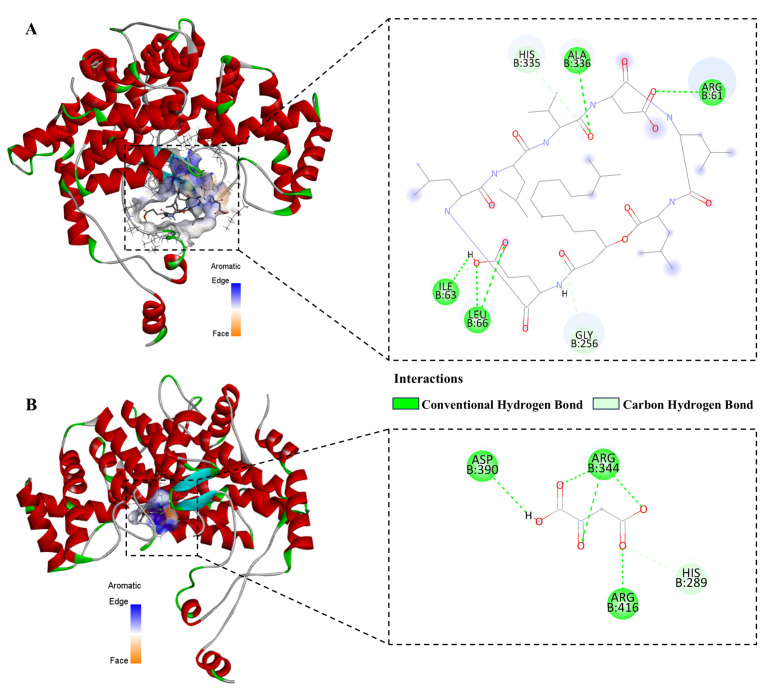
Docking map of surfactin, oxaloacetate, and citrate synthase protein. (**A**) The binding model of surfactin and citrate synthase. (**B**) The binding model of oxaloacetate and citrate synthase.

**Table 1 biomolecules-16-00051-t001:** Primers used for real-time PCR.

Gens	Gene Description	Primers (5′–3′)
g545	Vitamin D3 24-hydroxylase	CTTGACGACTCTCCCGAAGG
		GAAAAACCGACTCCCGTCCT
g77	Glycerol kinase	CAAGTCATTACGGGGCCTCA
		CAGAGTCGAGATGCCACCAA
g342	Glyceraldehyde dehydrogenase	CAAGTCCAGCAACGAGCAAA
		AAGGCCACCTTGTCTACGTC
g374	Homologous genes of killer toxin α/β subunits	TCCCGTGCCTCTTCTTTCAC
		CCCGGGTATTCCCAATCGAG
g332	Inositol polyphosphate kinase 2	ACGTGACCTGGATACCGAAC
		CCAGACCGATGCCACTATCA
ACT	Actin	TCGGTATGGGTCAGAAGGAC
		GGAGCCTCAGTCAACAGGAC

## Data Availability

The raw data of this study are publicly available at the China National Center for Bioinformation (CNCB) with the accession number CRA034366 (link: https://ngdc.cncb.ac.cn/gsub/submit/gsa/subCRA056264/finishedOverview, accessed on 30 November 2025).

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
