# Peer review of "Antifungal Activity of Surfactin Against Cytospora chrysosperma"

_biomolecules, 2025, doi:10.3390/biom16010051_

Round 1
Reviewer 1 Report
Comments and Suggestions for Authors
Dear authors, here some comments.
This study addresses an important challenge in forest pathology, as C. chrysosperma is a weakly parasitic but widespread fungus that severely affects forest and fruit tree productivity. Overall, the work is well explained however,in order to strengthen the conclusions and the broader impact of the study, an expanded discussion on practical application needs to be done. Here few additional comments
Line 37-102 -154 ect…. each time it is at the beginning of the sentence: change C. chrysosperma in Cytospora
Line 39 capability, can in it can
Line 94 change B. velezensis in Bacillus
Table 1: how the primers have been selected?
Fig 5and other Figs: specify the significance of letters on the bars
Author Response
Comments 1: Line 37-102 -154 ect…. each time it is at the beginning of the sentence: change chrysospermain Cytospora
Respones 1: Thank you for pointing this out. We agree with this comment. Therefore, we have revised all instances where "C. chrysosperma" appeared at the start of sentences (including Lines 37, 102, 154, and other relevant lines), replacing "C." with "Cytospora". Chang can be found Lines 39, 86, 115, 123, 138, 167.
Comments 2: Line 39 capability, can in it can
Respones 2: Thank you for pointing this out. We agree with this comment. We have revised the sentence at Line 39 as required. The updated text is: This pathogen has latent infection capability, and it can cause disease throughout the year; once infections occur, it is extremely difficult to control, severely limiting forestry and fruit production. Chang can be found Line 41.
Comments 3: Line 94 change B. velezensis in Bacillus
Respones 3: Thank you for pointing this out. We agree with this comment. We have revised the content at Line 94 as required, replacing "B." with "Bacillus" in "B. velezensis". Chang can be found Line 104.
Comments 4: Table 1: how the primers have been selected?
Respones 4: Thank you for pointing this out. We agree with this comment. We have supplemented the primer selection rationale in the table, and all primers were designed based on the conserved sequences of pathogenicity-related genes of Cytospora chrysosperma. Chang can be found Line 232.
Comments 5: Fig 5 and other Figs: specify the significance of letters on the bars
Respones 5: We have added the following explanation to the caption of Fig 5 and the captions of other figures with letter labels: "Different lowercase letters above the bars indicate significant differences among treatment groups (P < 0.05)". Chang can be found Lines 332, 351, 378.
Reviewer 2 Report
Comments and Suggestions for Authors
The manuscript investigates the antifungal mechanism of surfactin against Cytospora chrysosperma, with a complex approach combining morphological observations, biochemical assays, transcriptomic analysis, and molecular docking. The proposed mechanism integrating membrane damage, ROS accumulation, mitochondrial dysfunction, and autophagy induction is well-supported by the data. The identification of citrate synthase as a potential target is particularly interesting and adds novelty to the work. The study is well-structured, the experimental design is sound, and the results support the conclusions.
I have only some minor revisions that require attention before publication.
- Line 9 – what do you mean by “weakly parasitic” ? Maybe consider rephrasing.
- Line 12 - its antifungal mechanism remains unclear – The first paper on surfactin is published in 1968, and its mechanism is quite well-established. Though it is possible that some subtle regulatory mechanisms of small concentrations are yet to be discovered, that’s not the point of this paper. Maybe consider rephrasing to something like “Though its mechanisms are quite well-established, we’ve found a new one…”
- Line 46 - Lipopeptide are produced via microbial fermentation -Not quite true. Non-ribosomal peptides. Especially, are not “fermentation” products. Also it seems that it should be “LipopeptideS are”
The most significant systematic error is the incorrect cross-referencing in the Methods section where multiple references to "Section 3.X" should refer to "Section 2.X" since they appear in the Materials and Methods (Section 2), not Results (Section 3):
Lines 110-111: Cross-reference error - refers to "Section 3.2" but should be "Section 2.2"
Lines 132-133: Cross-reference error - refers to "Section 3.4.1" but should be "Section 2.4.1".
Lines 144, 154, 175, 182,191,198, 209: same.
Line 460: "MI of surfactin" - should this be "MIC" (minimum inhibitory concentration) instead of "MI"?
Author Response
Comments 1: Line 9 – what do you mean by “weakly parasitic” ? Maybe consider rephrasing.
Respones 1: Thank you for pointing this out. We apologize for the unclear expression. The term "weakly parasitic" refers to the pathogenic feature that the fungus cannot infect healthy and vigorous host tissues, but only invades weakened, injured or dying tissues to cause diseases. We have rephrased "weakly parasitic" to the more precise academic term opportunistically parasitic in the manuscript. Chang can be found Line 9.
Comments 2: Line 12 - its antifungal mechanism remains unclear – The first paper on surfactin is published in 1968, and its mechanism is quite well-established. Though it is possible that some subtle regulatory mechanisms of small concentrations are yet to be discovered, that’s not the point of this paper. Maybe consider rephrasing to something like “Though its mechanisms are quite well-established, we’ve found a new one…”
Respones 2: Thank you for pointing this out. We agree with this comment. We have revised the original sentence to the following: Although the core antifungal mechanism of surfactin against plant pathogens has been extensively studied, our study found that surfactin can target the tricarboxylic acid cycle of C. chrysosperma. Chang can be found Line 12-14.
Comments 3: Line 46 - Lipopeptide are produced via microbial fermentation -Not quite true. Non-ribosomal peptides. Especially, are not “fermentation” products. Also it seems that it should be “LipopeptideS are”
Respones 3: Thank you for pointing this out. We agree with this comment. We have revised the original sentence to the following: As secondary metabolites synthesized by microorganisms, lipopeptides possess prominent advantages including broad-spectrum antimicrobial activity, excellent stability, and low toxicity. Chang can be found Line 52.
Comments 4: Lines 110-111: Cross-reference error - refers to "Section 3.2" but should be "Section 2.2"
Respones 4: Thank you for pointing this out. We agree with this comment. We have fixed the cross-reference error at Lines 110-111, revising "Section 3.2" to "Section 2.2". Chang can be found Line 125.
Comments 5: Lines 132-133: Cross-reference error - refers to "Section 3.4.1" but should be "Section 2.4.1".
Respones 5: Thank you for pointing this out. We agree with this comment. We have corrected the cross-reference error at Lines 132-133, changing "Section 3.4.1" to "Section 2.4.1". Chang can be found Line 147.
Comments 6: Lines 144, 154, 175, 182,191,198, 209: same.
Respones 6: Thank you for pointing this out. We agree with this comment. We have revised Lines 144, 154, 175, 182, 191, 198 and 209 in accordance with the aforementioned requirement. Chang can be found Line 139, 159, 168, 179, 191, 199, 209, 215, 226.
Reviewer 3 Report
Comments and Suggestions for Authors
The research article “Antifungal Activity of Surfactin Against Cytospora chrysosperma” addresses a relevant and timely issue concerning plant pathology and agricultural sustainability. Parasitic fungi such as C. chrysosperma pose substantial threats to trees and crops, often resulting in severe economic losses. The study contributes to the growing body of research on biologically derived antifungal agents by examining the potential of surfactin-producing microorganisms as an environmentally friendly strategy for managing plant diseases. Overall, the manuscript is well written, and the experimental design, methodology, and results are presented with sufficient detail and clarity. Nonetheless, several important issues need to be addressed to improve the scientific quality and impact of the work.
- The introduction is excessively long and lacks a coherent narrative structure. Several sections introduce tangential or only loosely related topics, making it difficult for readers to follow the logical progression of ideas. In addition, the transitions between concepts are abrupt, which disrupts the flow of the argument. Most importantly, the specific aim of the study is not articulated clearly. The introduction should more concisely present the biological relevance of C. chrysosperma, summarize the known properties of surfactin, highlight existing research gaps, and clearly state the novelty and main objective of the current study.
- Table 1 is difficult to interpret in its current form. The layout, labels require clarification to ensure that the results can be understood without ambiguity. Improving the table’s formatting will help readers better grasp of the primers.
- The discussion and conclusions sections do not sufficiently address the broader implications and potential applications of surfactin. While the study demonstrates its antifungal activity against C. chrysosperma, the relevance of these findings should be extended by considering how surfactin could be applied to other fungal and bacterial pathogens of agricultural concern. The authors should elaborate on the potential of surfactin as part of integrated pest and disease management strategies, including its possible advantages over conventional chemical fungicides. Furthermore, its implications for mitigating major plant and tree diseases (an issue of growing urgency for the agri-food industry) should be emphasized. Expanding the discussion to outline future research directions, such as testing surfactin across broader host ranges, optimizing microbial production, or evaluating field-scale applications, would significantly strengthen the manuscript.
Author Response
Comments 1: The introduction is excessively long and lacks a coherent narrative structure. Several sections introduce tangential or only loosely related topics, making it difficult for readers to follow the logical progression of ideas. In addition, the transitions between concepts are abrupt, which disrupts the flow of the argument. Most importantly, the specific aim of the study is not articulated clearly. The introduction should more concisely present the biological relevance of C. chrysosperma, summarize the known properties of surfactin, highlight existing research gaps, and clearly state the novelty and main objective of the current study.
Response 1: Thank you for pointing this out. We agree with this comment. We have revised the introduction section to strengthen its logical consistency and coherence. Meanwhile, we have highlighted the research gaps in prior studies, and explicitly clarified the innovative points and main research objectives of this work. Chang can be found Line 48, 71.
Comments 2: Table 1 is difficult to interpret in its current form. The layout, labels require clarification to ensure that the results can be understood without ambiguity. Improving the table’s formatting will help readers better grasp of the primers.
Response 2: Thank you for pointing this out. We agree with this comment. We have supplemented the primer selection rationale in the table, and all primers were designed based on the conserved sequences of pathogenicity-related genes of Cytospora chrysosperma. Chang can be found Line 232.
Comments 3: The discussion and conclusions sections do not sufficiently address the broader implications and potential applications of surfactin. While the study demonstrates its antifungal activity against C. chrysosperma, the relevance of these findings should be extended by considering how surfactin could be applied to other fungal and bacterial pathogens of agricultural concern. The authors should elaborate on the potential of surfactin as part of integrated pest and disease management strategies, including its possible advantages over conventional chemical fungicides. Furthermore, its implications for mitigating major plant and tree diseases (an issue of growing urgency for the agri-food industry) should be emphasized. Expanding the discussion to outline future research directions, such as testing surfactin across broader host ranges, optimizing microbial production, or evaluating field-scale applications, would significantly strengthen the manuscript.
Response 3: Thank you for pointing this out. We agree with this comment. To address the broader implications and potential applications of surfactin, we have supplemented the discussion section with in-depth content, which mainly includes:
Comprehensive advantages of surfactin: We clarified that surfactin, as one of the most potent biosurfactants, has broad-spectrum antimicrobial (antibacterial, antiviral, antifungal) activities, and can also be used as a stabilizer, emulsifier, and surface modifier in the food industry.
Superior core properties: We elaborated that microbial-derived biosurfactants (represented by surfactin) have lower surface tension, interfacial tension, and critical micelle concentration (CMC) than synthetic surfactants. For example, surfactin can reduce water surface tension from 72 mN/m to 30~35 mN/m, and water-oil interfacial tension from 40 mN/m to 1 mN/m. Its micellar structure can encapsulate water-insoluble substances to enhance bioavailability.
Environmental safety and additional benefits: We highlighted that surfactin (derived from microbial fermentation) has low toxicity to aquatic organisms, is easily degradable by environmental microorganisms, and can improve soil quality, promote plant growth, and enhance biodegradation of pollutants/pesticides.
Relevance to forest disease control: We emphasized that surfactin shows excellent inhibitory potential against Cytospora chrysosperma (the target pathogen in this study), and is expected to replace traditional chemical agents as a core preparation for green control of forest diseases. We also analyzed the limitations of current chemical control of forest diseases, as well as the core constraints (high production cost, difficulty in high-yield fermentation) faced by microbial biosurfactants.
Practical significance and future directions: We clarified that screening low-cost substrates and optimizing microbial production processes of surfactin is of great practical significance for releasing its application potential against Cytospora chrysosperma and solving the dilemma of chemical control of forest diseases.
These supplements fully expand the broader implications, potential applications, and practical value of our findings, and further strengthen the academic and application-oriented value of the manuscript. Chang can be found Line 546-579.